# Temperature Rise Inside Shear Bands in a Simple Model Glass

**DOI:** 10.3390/ijms232012159

**Published:** 2022-10-12

**Authors:** Alexandra E. Lagogianni, Fathollah Varnik

**Affiliations:** Interdisciplinary Centre for Advanced Materials Simulation (ICAMS), Ruhr-University Bochum, Universitätsstraße 150, 44801 Bochum, Germany

**Keywords:** metallic glass, shear banding, plastic deformation, viscous heating, molecular dynamics simulation

## Abstract

One of the key factors, which hampers the application of metallic glasses as structural components, is the localization of deformation in narrow bands of a few tens up to one hundred nanometers thickness, the so-called shear bands. Processes, which occur inside shear bands are of central importance for the question whether a catastrophic failure of the material is unavoidable or can be circumvented or, at least, delayed. Via molecular dynamics simulations, this study addresses one of these processes, namely the local temperature rise due to viscous heat generation. The major contribution to energy dissipation is traced back to the plastic work performed by shear stress during steady deformation. Zones of largest strain contribute the most to this process and coincide with high-temperature domains (hottest spots) inside the sample. Magnitude of temperature rise can reach a few percent of the sample’s glass transition temperature. Consequences of these observations are discussed in the context of the current research in the field.

## 1. Introduction

It is well-known that brittleness is one of the main issues when it comes to the use of amorphous solids [1,2] as structural components. This problem is most prominent in metallic glasses, which, despite their excellent mechanical properties with regard to yield strength and toughness, cannot be deformed into a desired shape without the risk of failure [3,4,5]. This process is accompanied by the localization of deformation in narrow shear bands (SBs) with a typical thickness of a few tens of nanometers [6,7,8]. Despite their narrowness, SBs are known to influence the surrounding matrix in a relatively wide zone that extends up to hundreds of micrometers [9,10,11].

A question of central interest regarding the role and consequences of shear band formation in amorphous solids concerns the possibility of a local softening due to generation of heat, the latter arising from the work performed by the stress, $\dot{q}=\sigma \dot{\epsilon }$, where $\dot{q}$ is the local rate of heat creation per unit volume, σ is the locally acting shear stress and $\dot{\epsilon }$ the local shear rate. Such a heat-related softening could result if the rate of heat generation is sufficiently high in order to induce a noticeable temperature rise, ΔT. Depending on the *T*-range and magnitude of ΔT, heating a metallic glass may also have the opposite effect of annealing, an acceleration of relaxation processes, which often leads to a more brittle state.

However, it is rather difficult to experimentally and accurately access heat generation and temperature inside the bulk of material (i.e, away from surfaces), and one is thus often restricted to apply indirect methods and rough estimates. Indeed, reported values of ΔT in the existing literature are distributed over a broad range, differing sometimes by several orders of magnitude [7,12,13,14,15,16,17].

In this context, molecular dynamics simulations present a promising alternative as they provide access to a fully detailed information regarding positions and momenta of all particles and forces acting on each individual particle. From this set of fundamental data, physically important derived quantities can be constructed on the local scale, i.e., around a small region in space or in the direct neighborhood of a particle. Important examples of such quantities, which are also directly relevant for the present study, include strain, stress, energy production rate and temperature.

The present work sets on at this point. Building upon our ample experience with properties of the so-called Kob–Anderson binary Lennard–Jones mixture [18,19] as a generic glass former, we perform molecular dynamics simulations under a set of parameters for which we already know that strain has the strong tendency of localization and the formation of system spanning shear bands under an externally imposed deformation [20,21,22]. Moreover, even though shear bands may also occur upon periodic boundary conditions using SLLOD-equations [23,24], we use amorphous frozen walls to stabilize SBs, thus simplifying a statistical analysis of their properties [21].

All the above studies have been performed at imposed temperature conditions, i.e., by allowing the exchange of energy with a heat reservoir, the latter being coupled to all the particles. In this work, however, we do not impose temperature inside the bulk of the sample since we aim at studying how it changes due to the generation of heat within shear bands. Instead, we mimic experimental conditions, where external boundaries are at a constant temperature. This leads to a temperature gradient between the inner part and boundaries, such that the energy flux towards boundaries carries away the heat generated by viscous dissipation. A similar procedure has been used recently to study traction, non-equilibrium phase behavior and heating under shear at high pressures in the context of tribology [25,26].

As will be shown below, we provide direct numerical evidence for the generation of heat inside shear bands and for the resulting temperature increase. Plastic deformation, the local heat generation rate, $\dot{q}$, and the resulting rise in temperature closely follow the path of shear band along the sample. Maxima of temperature profiles (determined along the direction perpendicular to shear band) coincide with the highest $\dot{q}$ and largest strain. The magnitude of temperature rise observed in our simulations can reach roughly five percent of the glass transition temperature, $T_{g}$. Noting that the $T_{g}$ of Zr-based and PdNi-based bulk metallic glasses lie in the range of 500 °K–700 °K [27,28], this would amount to a temperature variation of roughly a few tens of Kelvin, Δ*T* ≈ 20 °K–40 °K. Even though not too large, such a temperature increase may have important consequences for relaxation processes depending on the operating temperature [27].

## 2. Results

For the present study, four statistically independent samples were generated. A simple way to achieve this is to use, for each initial configuration, a Maxwell–Boltzmann velocity distribution with a different seed for random number generator. The above described preparation protocol is then applied to these samples. Under shear, all these four configurations exhibit system spanning SBs. Since all these samples show qualitatively similar behavior, we will focus in the following on one of them as a representative for all others.

Since the spatial organization of shear deformation plays a fundamental role in this work, we first address this issue. Figure 1a illustrates the strain map for the xz-component of the strain tensor, $\epsilon_{xz}$, at various instants of shear deformation. The corresponding stress-vs.-strain curve is plotted in Figure 1b. As seen from this figure, at small global strains belonging to the elastic response regime, deformation is homogeneously distributed inside the simulation box. As the imposed strain approaches a yielding threshold ($\epsilon_{xz}^{\mathrm{yield}}\approx 0.08$, Figure 1b), where stress attains a maximum, the strain map clearly shows the formation of a narrow system spanning shear band. For strains beyond this plastic threshold, the SB changes to some extent, but its overall propagation direction does not alter (see also a discussion of Figure 2 below).

The strain maps shown in Figure 1 are all determined using particle positions between two instants in time. This is achieved via a well-known coarse-graining procedure [29,30], which allows us to define a local deformation tensor, based on displacements of particles in the neighborhood of a given point in space. First, a coarse-grained displacement-field, u(r), is calculated via
(1)$$\begin{array}{c} u\left(r,t,\Delta t\right)=\frac{\sum_{i}u_{i}\left(t,t+\Delta t\right)\;\phi \left(r-r_{i}\left(t\right)\right)}{\sum_{j}\phi \left(r-r_{j}\left(t\right)\right)}\;, \end{array}$$
where $u_{i}\left(t,\Delta t\right)=r_{i}\left(t+\Delta t\right)-r_{i}\left(t\right)$ is the displacement of particle *i* between times *t* and t+Δt. In Equation (Equation 1), $\phi \left(r\right)=\frac{1}{{\left(4\pi w^{2}\right)}^{3/2}}\exp\left(-\frac{r^{2}}{2w^{2}}\right)$ is a Gaussian coarse-graining function with *w* denoting the coarse-graining scale and *r* the absolute magnitude of the position vector r. For all calculations reported here, a coarse-graining length of w=2.5 was used. This distance corresponds roughly to the position of the second minimum of the radial distribution function [19]. We evaluate u(r) on a simple-cubic lattice with a lattice constant of a=2 particle diameters. For the time interval, Δt, we chose a value which corresponds to 1% overall imposed deformation, i.e., $\Delta t=0.01/{\dot{\epsilon }}_{xz}=100$ (in units of $t_{\mathrm{LJ}}$), where we used ${\dot{\epsilon }}_{xz}=10^{-4}$. Using the thus obtained coarse-grained displacement field, and assuming small deformation, we calculate the local strain tensor via
(2)$$\begin{array}{c} \epsilon_{\alpha \beta }\left(r\right)=\frac{1}{2}\left[\frac{\partial u_{\alpha }\left(r\right)}{\partial x_{\beta }}+\frac{\partial u_{\beta }\left(r\right)}{\partial x_{\alpha }}\right], \end{array}$$
with $x_{\alpha },x_{\beta }\in \left\{x,\;y,\;z\right\}$, $u_{\alpha },u_{\beta }\in \left\{u_{x},\;u_{y}\;u_{z}\right\}$.

To evaluate temperature within the shear band, we first introduce a criterion, which allows us to identify particles belonging to the SB. Guided by the fact that particles within the shear band contribute more than the average value to the yielding threshold (to compensate the smaller contribution of those which lie outside SB), and at the same time trying not to exclude too many particles from the analysis (in order to keep statistical noise as small as possible), we assign a particle *i* to the shear band if the coarse-grained strain Equation (Equation 2) at the position of that particle satisfies the following condition: $\epsilon_{xz}\left(r_{i}\right)\geq 0.1$ (see also Figure S2 in [21]). Using this criterion, we can determine the local temperature within the shear band as the average kinetic energy over all the particles assigned to the SB,
(3)$$\begin{array}{c} T_{\mathrm{SB}}=\frac{m}{k_{B}N_{\mathrm{SB}}}\sum_{i\in \mathrm{SB}}v_{y,i}^{2}, \end{array}$$
where $N_{\mathrm{SB}}$ is the number of particles inside (or assigned to) SB. Importantly, in Equation (Equation 3), we only consider a particle velocity component along the so-called vorticity direction (which turns out to be the *y*-component here). By doing so, we avoid possible complications due to the strain in the xz-plane.

The thus obtained shear-band-temperature, $T_{\mathrm{SB}}$, is shown in Figure 1b together with its system-averaged counterpart (indicated as ‘Whole system’). Both these temperatures are essentially constant at strains below the yielding threshold but rise beyond this limit, reaching a plateau at large strains, where the shear stress also reaches a roughly constant value. The delay between shear stress and temperature is in line with the idea that the early elastic part of deformation stores the mechanical energy and thus has no contribution to heat dissipation. The latter starts only as plastic deformation becomes important, which is certainly the case for strains beyond the yielding point.

The shear band observed in the present set of simulations is essentially aligned along the horizontal *x*-axis, and the gradient direction is along the *z*-axis. This motivates us to visualize the path of the shear band within the xz-plane. For this purpose, we divide the system in slabs of roughly five particle thickness, Δx=5, and, within each slab, perform an averaging over the positions of only those particles inside the slab, which belong to the shear band. Via this procedure, we generate pairs of numbers ($x_{\mathrm{SB}},z_{\mathrm{SB}}$), specifying the geometric centers (centroid positions) of the shear band within the considered slabs. Note that the horizontal SB-coordinate, $x_{\mathrm{SB}}$, is not necessarily equal to the lateral coordinate associated with the midpoint of the slab. A distinction between these two quantities, however, is irrelevant for the present analysis, since, as we have verified, the difference between $x_{\mathrm{SB}}$ and $x_{\mathrm{slab}-\mathrm{middle}}$ is quite small in all the cases of interest to this study (data not shown). The use of the lateral SB-coordinate $x_{\mathrm{SB}}$ is just for the sake of consistency in using the results obtained by applying the above introduced averaging procedure.

This geometric representation of shear band is plotted in Figure 2 for three selected strain intervals as indicated in the curve legends. Each data set shown in this figure is obtained as an average over 10 sequential evaluations, each covering 1%-strain. As seen from this figure, the detailed path of shear band changes as deformation proceeds further. It is also noticeable that relatively sharp kinks which are visible in the SB-path right after the yielding threshold ($\Delta \epsilon_{xz}$ = 10–20%), disappear at later stages of deformation. This suggests that material has become softer than it was at the beginning of shear. This is also in-line with reports on mechanically induced rejuvenation of amorphous solids [31,32].

The close connection between temperature rise and shear bands is further highlighted in Figure 3, where for each set of data shown in Figure 2 the corresponding temperature along the respective SB-path is plotted. A comparison between these two figures reveals that temperature closely follows the SB-path and its gradients. The variations of temperature along the SB-path provide hints towards the possible role of structural heterogeneity in influencing heat generation and heat conduction inside shear band.

To proceed further, we subdivide the xz-plane in squares of size 5×5 without subdivision for the *y*-direction. This makes equally sized cuboids of linear dimensions 5×10×5. In this procedure, we consider only those cuboids whose *z*-positions satisfy $z_{\mathrm{cuboid}}\in \left[-47.5,\;47.5\right]$. This serves to eliminate undesired wall effects, which, while interesting on their own right, would bias the bulk-properties we are aiming to study here. Using the thus generated grid, we determine temperature, strain, stress and heat generation rate within each individual cuboid. The thus obtained results for temperature field are shown as a color map in Figure 4 along with the SB-path, the latter obtained as geometric center (see Figure 2). Figure 4 clearly supports the idea that high temperatures always occur within the shear band and thereby provide a genuine evidence on heat generation and its impact on temperature inside the shear band.

Next, we work out a simple relation between the imposed strain rate, heat generation rate and temperature, which proves quite useful for an understanding of the obtained simulation results. For this purpose, we start with the well known relation $\dot{q}=\sigma_{xz}{\dot{\epsilon }}_{xz}^{\mathrm{plastic}}$, where $\dot{q}$ is the amount of generated heat per unit time and unit volume, and ${\dot{\epsilon }}_{xz}^{\mathrm{plastic}}$ is the rate of plastic deformation. The latter can be expressed in terms of the globally imposed strain rate and the elastic contribution to deformation, ${\dot{\epsilon }}_{xz}^{\mathrm{plastic}}={\dot{\epsilon }}_{xz}-{\dot{\epsilon }}_{xz}^{\mathrm{elastic}}$. The elastic strain can be roughly approximated as $\epsilon_{xz}^{\mathrm{elastic}}\approx \sigma_{xz}/G$, where *G* is the elastic shear modulus, i.e., the slope of the stress–strain curve at small strains. Using this relation, one arrives at ${\dot{\epsilon }}_{xz}^{\mathrm{plastic}}\approx {\dot{\epsilon }}_{xz}-{\dot{\sigma }}_{xz}/G$.

Importantly, the elastic contribution, ${\dot{\sigma }}_{xz}/G$, rapidly decays as stress reaches a roughly time independent value. This occurs for globally imposed strains, $\epsilon_{xz}\geq 0.2$, see Figure 1b. In this case, ${\dot{\epsilon }}_{xz}^{\mathrm{plastic}}\approx {\dot{\epsilon }}_{xz}$, and one finally obtains for the amount of heat produced per unit time and unit volume the simple relation, $\dot{q}\approx \sigma_{xz}{\dot{\epsilon }}_{xz}\approx$ constant in the large strain limit. It then follows from the energy balance equation, $\dot{q}=-\nabla \cdot j_{q}$, ($j_{q}$ is the heat flux density) and symmetry properties of the present slab geometry that $\frac{\partial j_{z,q}}{\partial z}=$ constant. When combined with Fourier’s law, $j_{z,q}=-\lambda_{T}\frac{\partial T}{\partial z}$, one arrives at $\frac{\partial^{2}T}{\partial z^{2}}=$ constant, where we assumed that the coefficient of heat conductivity, $\lambda_{T}$, does not change across the shear band. A direct consequence of this simplified model is that the graph of T(z) should describe a parabola in the domain, where heat is generated, i.e., within the shear band. Beyond SB, T(z) would approximately follow a straight line, since $\dot{q}=0$ and thus $j_{z,q}=$ constant in this region. Data shown in Figure 5 are qualitatively in line with this simple picture.

A further interesting observation based on the above data is that maxima of temperature profiles approximately coincide with geometric centers along the SB-path (Figure 4). This suggests that the location of the shear band can be estimated by an analysis of high temperature zones inside the sample. This provides an alternative approach to access localized strain inside the material. This issue is more closely examined in Figure 6, where profiles of temperature, *T*, heat generation rate, $\dot{q}$, and shear strain, $\epsilon_{xz}$, are plotted versus vertical coordinate, *z*. As seen from this figure, the maximum temperatures occur roughly at the same position as the shear strain, and $\dot{q}$ develop their corresponding peaks. Even though intuitively appealing, such an observation is highly non-obvious as heterogeneity of the structure can in principle lead to other non-trivial scenarios.

## 3. Materials and Methods

We perform non-equilibrium molecular dynamics simulations [33] of the well-known binary Lennard–Jones (LJ) mixture, first explored in a series of seminal papers by Kob and Andersen [18,19] and Kob and Barrat [34,35]. The model consists of two particle types, called A and B, with the number of fractions 80% (A particles) and 20% (B particles). Both particle types have equal masses, $m_{A}=m_{B}=1$. During all the simulations reported below, volume and particle number are kept constant. The total number density is $\rho =\rho_{A}+\rho_{B}=1.2$. Since the mass of both particle types is set to unity, we do not distinguish between mass and number densities in this study.

The particles interact via the 6-12 Lennard–Jones (LJ) potential defined as $U_{\mathrm{LJ}}=4\epsilon_{\alpha \beta }\left[{\left(d_{\alpha \beta }/r\right)}^{12}-{\left(d_{\alpha \beta }/r\right)}^{6}\right]$, with α,β∈{A,B}. The parameters of the model are set to $\epsilon_{\mathrm{AA}}=1$, $\epsilon_{\mathrm{AB}}$ = 1.5$\epsilon_{\mathrm{AA}}$, $\epsilon_{\mathrm{BB}}$ = 0.5$\epsilon_{\mathrm{AA}}$, $d_{\mathrm{AB}}$ = 0.8$d_{\mathrm{AA}}$, $d_{\mathrm{BB}}=0.88d_{\mathrm{AA}}$.

The units of energy, length and mass are chosen to be $\epsilon_{\mathrm{AA}}$, $d_{\mathrm{AA}}$ and $m_{A}$. All other units are derived as combinations of these three basic units. The unit of time, for example, is given by $t_{\mathrm{LJ}}=d_{\mathrm{AA}}\sqrt{\frac{m_{A}}{\epsilon_{\mathrm{AA}}}}$. Temperature is measured in units of $\epsilon_{\mathrm{AA}}/k_{B}$, with Boltzmann constant, $k_{B}=1$. For the sake of computational performance and also for the reason of comparability with results obtained within previous simulation studies, the interaction potential is truncated at a cut-off distance of $r_{c,\alpha \beta }$ = 2.245$d_{\alpha \beta }$.

The time evolution of the system is achieved by solving Newtonian equations of motion via the velocity-Verlet algorithm with an integration time step of dt = 0.008. All the quantities are given in the above described LJ units.

The glassy configurations studied here are prepared following a protocol described in [21,36]. First, a simulation box of linear dimensions $L_{x}\times L_{y}\times L_{z}=2000\times 10\times 106$ containing a total of $N=2.544\times 10^{6}$ particles is simulated at a temperature of T=2, corresponding to the liquid state of the model (we recall here that the glass transition temperature of the model is roughly $T_{g}=0.4$ [32,37]). To ensure a constant temperature, particle velocities are coupled to a heat bath via the Nosé–Hoover thermostat [38,39]. These liquid-phase simulations are performed for a sufficiently long time so that particles can move distances roughly ten times their own diameter. In this stage, dynamic and static quantities are monitored to ensure that the system is well equilibrated, in the sense that properties become independent of the instant in time, where a measurement is started (time translation invariance). The system is then cooled rapidly to a temperature of T=0.2 belonging to the glassy state, and the system dynamics are allowed to evolve for a time span of $4\times 10^{4}t_{\mathrm{LJ}}$. During this physical aging step, the system evolves towards a practically unreachable equilibrium state, and its structure becomes closer to that of an amorphous solid [37]. After this, a second cooling stage is started during which all particles are coupled to a drag force, proportional to their respective velocities. The proportionality factor of this force is chosen to be constant and equal for all particles. As time proceeds, the system loses energy due to these over-damped Langevin dynamics and visits progressively lower energies. This represents the final step in the sample preparation process and is stopped as soon as the system’s temperature reaches $T=10^{-4}$. As examined by a survey of dynamic correlations, for the present model, this temperature corresponds to a quasi-athermal limit [20] in the sense that all types of structural relaxation processes are frozen, and only short time vibrational dynamics are present.

Next, two xy-layers, each with a thickness of three particle diameters are selected, one at the bottom (i.e., containing particles, *i*, whose centers lie between $-53<z_{i}<-50$) and one on the top ($50<z_{i}<53$) of the simulation box. These layers serve as confining walls. In addition to interactions with neighboring particles, each ‘wall particle’ is connected to a lattice site via a harmonic spring. Lattice sites are generated by duplicating the positions of wall particles. Thus, at t=0, no force is exerted by a lattice site upon its corresponding wall particle due to the zero distance between the two. A shear is imposed on the system by moving (along the positive *x*-direction) all the lattice sites of the upper wall with a constant velocity of $U_{\mathrm{top}-\mathrm{wall}}=\frac{1}{2}\left(L_{z}-6\right){\dot{\epsilon }}_{xz}=50{\dot{\epsilon }}_{xz}=5\times 10^{-3}$, where ${\dot{\epsilon }}_{xz}=10^{-4}$ is the global shear rate. Simultaneously, the lattice sites on the bottom are moved with the velocity $U_{\mathrm{bottom}-\mathrm{wall}}=-U_{\mathrm{top}-\mathrm{wall}}$. Since each wall is connected to its corresponding lattice via harmonic springs, it moves on average with the lattice velocity. The thus realized relative motion of the upper and lower walls induces a shear deformation in the system.

A technical note regarding the motion of walls is necessary here. If we impose a velocity on the walls by moving all the wall atoms exactly with the same constant velocity, wall particles would not have a thermal motion as there would be no fluctuations in their velocities. However, since the goal is to impose a constant temperature to the walls, thermal motion of wall atoms becomes important and cannot be left out [40,41,42].

The temperature of the two walls is chosen to be equal and is kept at a constant value of $T=10^{-4}$ by applying the Nosé–Hoover thermostat [38,39]. Particles between the walls obey Newtonian dynamics without any coupling to a thermostat. The temperature of the bulk thus results from exchange of energy with the walls. This setup allows us to address the effect of viscous heat generation due to the shear transformation events inside the system. As will be shown below, the major part of heat production is concentrated inside the SB, i.e, within a system spanning a narrow region, where the shear deformation is localized.

All the simulations performed in this study were conducted using the massively parallel molecular dynamics simulation software LAMMPS [43]. The 3D visualization and the color coding was performed by the OVITO software [44].

## 4. Conclusions and Outlook

This work presents results of molecular dynamic simulations of a model glass (the well-known Kob–Andersen binary Lennard–Jones mixture) on generation of viscous heat in shear bands during mechanical deformation. As described in the introduction, this model has been demonstrated to reproduce various features of bulk metallic glasses. Therefore, the present study is relevant for a better understanding of plasticity and heat generation, and the resulting consequences thereof on properties (e.g., ductility) of the material inside shear bands.

For the sake of this study, shear bands have been stabilized by introducing two planar walls with the same amorphous structure as the material itself. Using the thus designed setup, simulations were performed under steady shear. It is shown that temperature inside the sample first remains essentially constant in the early elastic deformation regime but then starts to rise as soon as plastic deformation begins. A further analysis reveals that this temperature increase is far stronger (roughly by a factor of five in our study) inside the shear band as compared to the matrix outside. A temperature gradient thus forms between the shear band and the outer matrix (defined as the region outside the shear band excluding walls), which remains essentially undeformed. In view of the importance of temperature for amorphous solids in general, and especially for mechanical properties of bulk metallic glasses, the present findings call for additional studies to explore consequences of this observation.

We also note that the largest temperature variations observed in our simulations were roughly five percent of $T_{g}$ and can probably not lead to a softening of mechanical properties if shear banding occurs as cryogenic temperatures, i.e., at *T* far below $T_{g}$. At temperature below but not too far from $T_{g}$, however, a rise in *T* by $\Delta T\approx 0.05T_{g}$ may trigger relaxation processes and lead to accelerated annealing.

New questions emerge from our studies, the most important of which is probably related to ductility. An important issue here regards competing processes which occur simultaneously upon mechanical deformation, e.g., inherent structural relaxation and plastic rearrangements on the one hand and shear-driven processes which drive the system out of equilibrium on the other hand. What is the outcome of these processes? Does the material soften under steady deformation inside shear bands or does it undergo accelerated physical aging and thus become more brittle due to elevated temperatures? What is the role of structure and its spatial heterogeneity here? While the present study provides direct numerical evidence for the significance of heat generation inside shear bands, these questions clearly underline the need for future studies.

## Figures and Tables

**Figure 1 ijms-23-12159-f001:**
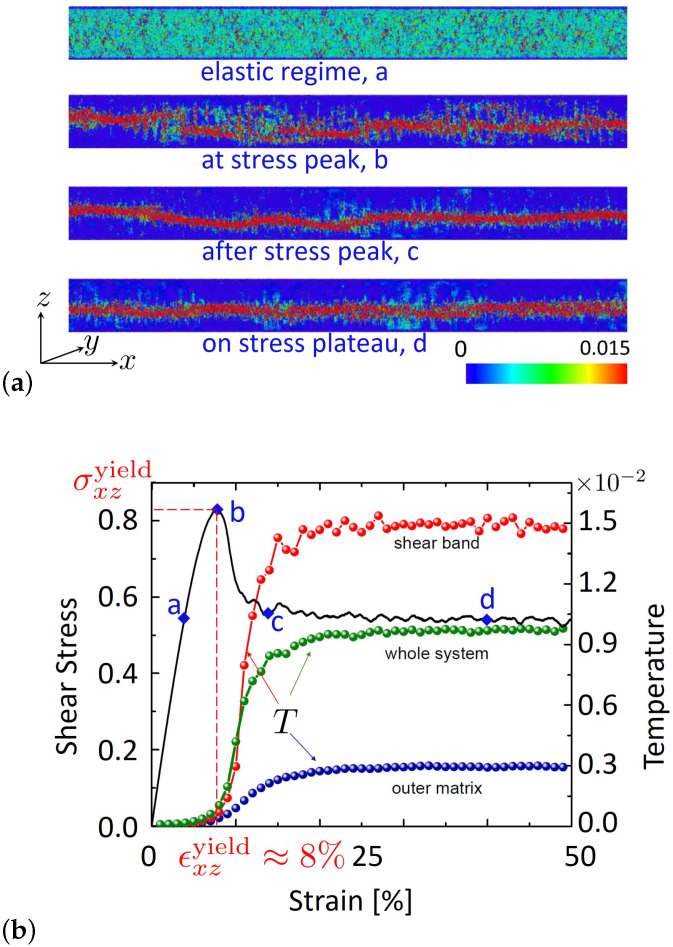
(**a**) Snapshots of the spontaneous shear strain (obtained during a time interval corresponding to 1% imposed global strain) during characteristic stages of shear deformation. The local shear strain first builds up homogeneously during the early elastic regime, where shear stress linearly grows with the externally imposed strain. A shear band is then formed as the stress reaches a maximum value. The shape of this SB slightly fluctuates during further stages of deformation (c, d), but its overall structure remains unchanged. The stress states corresponding to these snapshots are marked by blue diamonds, and the letters a, b, c and d on the stress-versus-strain plot in panel (**b**). The color bar gives the strain magnitude. These images (adapted) are taken from [21]. (**b**) The evolution of system averaged shear stress and temperature in the course of deformation is shown. For *T*, the corresponding data within the shear band and in the outer matrix (the region outside the shear band excluding walls) are also shown. The observation of a higher temperature inside SB is in line with the idea that heat is primarily generated there as compared to other parts of the sample. Note that *T* varies roughly by a factor of five between SB and the outer matrix.

**Figure 2 ijms-23-12159-f002:**
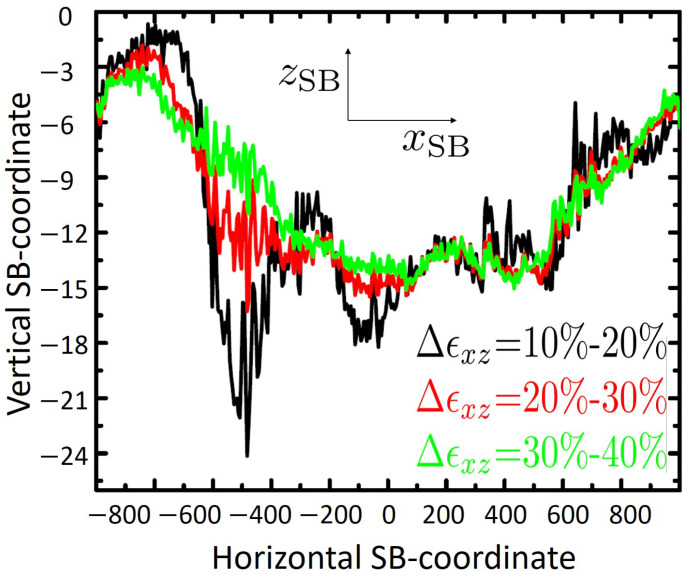
Average vertical coordinate of shear band, $z_{\mathrm{SB}}$, versus lateral position, $x_{\mathrm{SB}}$, evaluated for three different shear strain intervals as indicated. Each pair of $x_{\mathrm{SB}}$ and $z_{\mathrm{SB}}$ numbers is obtained as geometric center of particles within a slab of thickness Δx=5 which belong to the shear band. Each curve shown here represents an average over ten calculations, performed for small strain intervals of $\left[\epsilon_{xz},\epsilon_{xz}+1\%\right]$, $\left[\epsilon_{xz}+1\%,\epsilon_{xz}+2\%\right]$, …, $\left[\epsilon_{xz}+9\%,\epsilon_{xz}+10\%\right]$ with $\epsilon_{xz}=10\%$, 20% and 30%, respectively.

**Figure 3 ijms-23-12159-f003:**
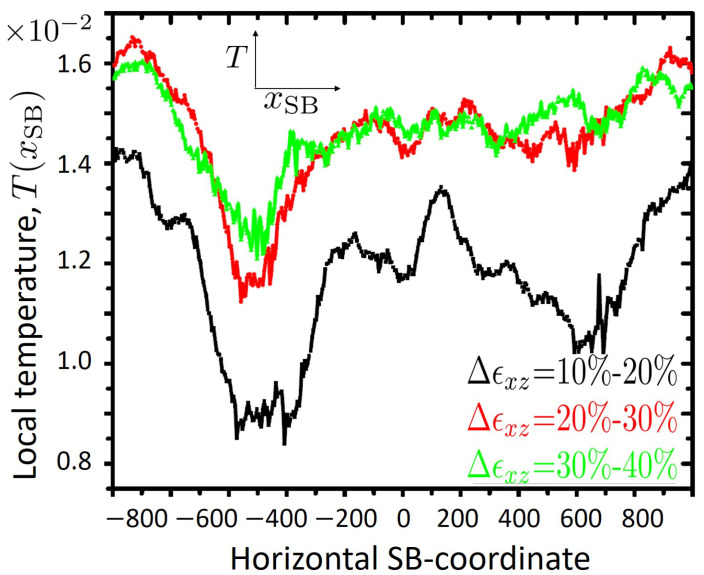
Temperature versus $x_{\mathrm{SB}}$ (see Figure 2), evaluated for three different shear strain intervals as indicated. The horizontal SB-coordinate is obtained as the *x*-component of the geometric center of particles within a slab of thickness Δx=5 which belong to the shear band. It essentially coincides with the midpoint of the slab. Temperature is obtained from the average kinetic energy of those particles which are considered when calculating $x_{\mathrm{SB}}$. Each curve shown here represents an average over ten calculations, performed for small strain intervals of $\left[\epsilon_{xz},\epsilon_{xz}+1\%\right]$, $\left[\epsilon_{xz}+1\%,\epsilon_{xz}+2\%\right]$, …, $\left[\epsilon_{xz}+9\%,\epsilon_{xz}+10\%\right]$ with $\epsilon_{xz}=10\%$, 20% and 30%, respectively.

**Figure 4 ijms-23-12159-f004:**
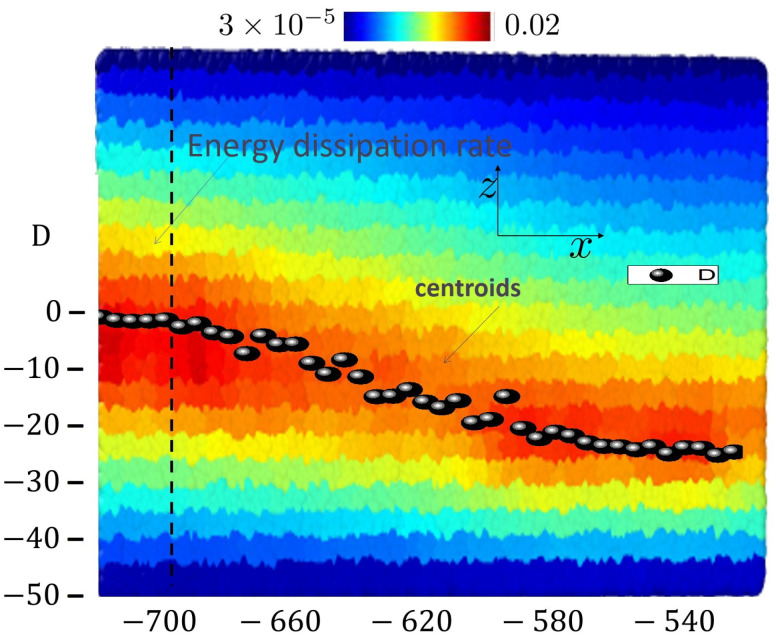
Color coded temperature field at a global strain of $\epsilon_{xz}=15$%, corresponding to a time of $t=0.15/{\dot{\epsilon }}_{xz}=1500t_{\mathrm{LJ}}$. The color bar shows the temperature range used in color coding. The ellipsoids (called in the image centroids) track the average position of the shear band. As expected, the highest temperatures occur inside the shear band. The dashed vertical line marks one of the cross sections for which temperature profiles are extracted from this data set and plotted in Figure 5, thus permitting a more quantitative analysis.

**Figure 5 ijms-23-12159-f005:**
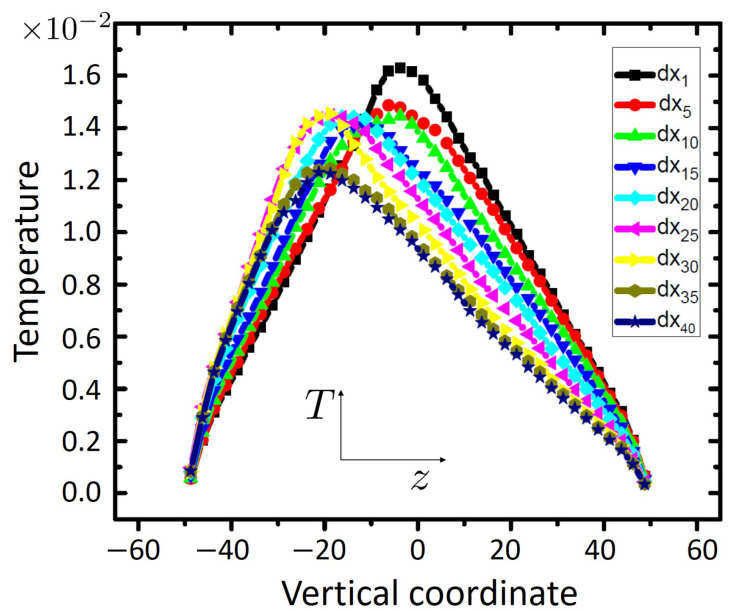
Temperature profiles (*T* versus the vertical coordinate, *z*) evaluated within thin parallel slabs of five particle diameter thickness. The curve legends give the integer index of the considered slab (e.g., $dx_{5}$ is the slab number 5 along the *x*-direction).

**Figure 6 ijms-23-12159-f006:**
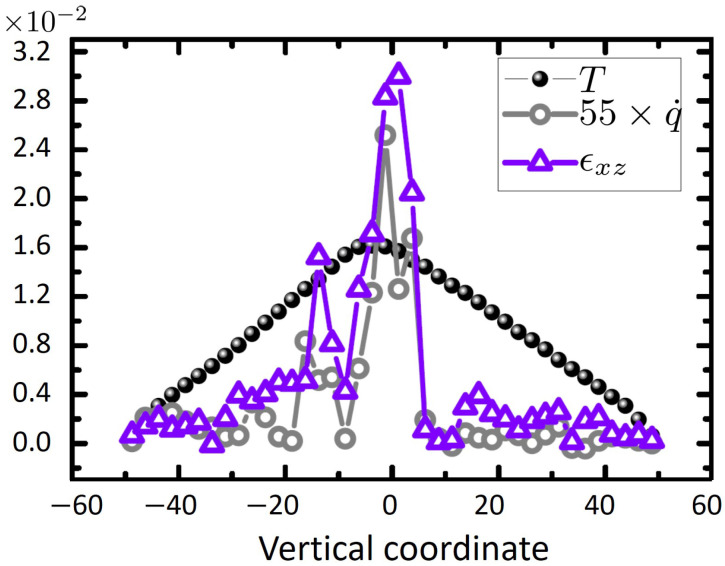
Temperature, local strain and heat production rate versus vertical coordinate *z*. As indicated in the curve legend, $\dot{q}$ is multiplied with a factor of 55 for better visibility. Within statistical fluctuations, the maxima of strain, heat production rate and temperature coincide, highlighting the close connection between these quantities. We have repeated this analysis for other slabs along the *x*-axis and observe the same trend (data not shown).

## Data Availability

Ruhr-University Bochum is currently setting up a research data management system. The present data will be saved there, as soon as the platform is operational. The expected time is March 2023. In the meanwhile, the data reported here are saved on an internal data storage system at ICAMS.

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
