# Peer review of "Temperature Rise Inside Shear Bands in a Simple Model Glass"

_ijms, 2022, doi:10.3390/ijms232012159_

Round 1

Reviewer 1 Report

attached

Reviewer 2 Report

This manuscript uses molecular dynamics simulations to study the local temperature rise due to viscous heat generation in a model glass during shear banding. The major contribution to energy dissipation is traced back to the plastic work done by shear stress during steady deformation. Zones of largest strain contribute the most to this process and coincide with high-temperature domains inside the sample. The magnitude of the temperature rise can reach a few percent of the sample’s glass transition temperature.

The manuscript is interesting and mostly well written. Overall, I recommend publication following the below minor clarifications.

1.       Typo (Title and Abstract: ‘Temperature raise inside shear bands in a simple model glass’ à ‘Temperature rise inside shear bands in a simple model glass’. In quite a few places in the manuscript, ‘raise’ should be replaced by ‘rise’.

2.       It is worth noting in the methodology that these are nonequilibrium molecular dynamics (NEMD) simulations, as proposed by Ashurst and Hoover.(1)

3.       In the methodology, it is worth clarifying that shear banding can only be captured by applying velocity to outer layer of atoms(2) rather than a constant velocity gradient as in other NEMD methods such as SLLOD.(3)

4.       It is not strictly true that different regions of the inhomogeneous system can be in different statistical mechanical ensembles (i.e. NVE and NVT). It is more accurate to just state which regions are thermostatted and which are not.

5.       The importance of not directly thermostatting the fluid atoms in confined NEMD simulations to enable viscous heating has been discussed previously.(4-6)

6.       The shape of the through-thickness temperature profiles due to shear heating have previously been investigated using NEMD simulations in glassy atomic(7) and molecular(8) systems during shear banding. The latter study also used Fourier’s law of heat conduction to calculate the thermal conductivity of the fluid from the curvature of the parabolic temperature profiles. The thermal conductivity values calculated with this method could be compared to previous thermal NEMD simulations of Kob-Andersen glasses.(9)

References

(1) Ashurst, W. T.; Hoover, W. G. Dense-Fluid Shear Viscosity via Nonequilibrium Molecular Dynamics. Phys Rev A 1975, 11, 658–678.

(2) Cao, J.; Likhtman, A. E. Shear Banding in Molecular Dynamics of Polymer Melts. Phys Rev Lett 2012, 108, 028302.

(3) Evans, D. J.; Morriss, G. P. Nonlinear-Response Theory for Steady Planar Couette Flow. Phys Rev A 1984, 30, 1528–1530.

(4) Khare, R.; de Pablo, J.; Yethiraj, A. Molecular Simulation and Continuum Mechanics Study of Simple Fluids in Non-Isothermal Planar Couette Flows. J Chem Phys 1997, 107, 2589.

(5) Bernardi, S.; Todd, B. D.; Searles, D. J. Thermostating Highly Confined Fluids. J Chem Phys 2010, 132, 244706.

(6) Yong, X.; Zhang, L. T. Thermostats and Thermostat Strategies for Molecular Dynamics Simulations of Nanofluidics. J Chem Phys 2013, 138, 084503.

(7) Gattinoni, C.; Heyes, D. M.; Lorenz, C. D.; Dini, D. Traction and Nonequilibrium Phase Behavior of Confined Sheared Liquids at High Pressure. Phys Rev E 2013, 88, 052406.

(8) Ewen, J. P.; Gao, H.; Müser, M. H.; Dini, D. Shear Heating, Flow, and Friction of Confined Molecular Fluids at High Pressure. Physical Chemistry Chemical Physics 2019, 21, 5813.

(9) Bhuyan, P. J.; Mandal, R.; Chaudhuri, P.; Dhar, A.; Dasgupta, C. Aging Effects on Thermal Conductivity of Glass-Forming Liquids. Phys Rev E 2020, 101. 

Reviewer 3 Report

The paper by Lagogianni and Varnik reports on a simulation study of viscous heat generation in shear bands during mechanical deformation. The work is correct and potentially interesting for the community working on plasticity in amorphous materials. The simulations are properly conducted and analyzed. The manuscript is well written: the presentation and discussion of results is clear and detailed. My only concern is related to the feeling I had that the present work is rather preliminary. It leaves some questions open, as mentioned by the authors themselves, which I expect will be addressed in future works.

In summary the paper can be published as it stands in IJMS. 
